# Effect of Various Plasticizers in Different Concentrations on Physical, Thermal, Mechanical, and Structural Properties of Wheat Starch-Based Films

**DOI:** 10.3390/polym15010063

**Published:** 2022-12-23

**Authors:** Abdulrahman A. B. A. Mohammed, Zaimah Hasan, Abdoulhdi A. Borhana Omran, Abdulhafid M. Elfaghi, M.A. Khattak, R. A. Ilyas, S. M. Sapuan

**Affiliations:** 1Institute of Sustainable Energy, Universiti Tenaga Nasional, Jalan Ikram-Uniten, Kajang 43000, Selangor, Malaysia; 2Department of Mechanical and Mechatronic Engineering, Faculty of Engineering, Sohar University, Sohar P C-311, Oman; 3Department of Mechanical Engineering, College of Engineering Science & Technology, Sebha University, Sabha 00218, Libya; 4Faculty of Mechanical and Manufacturing Engineering, Universiti Tun Hussein Onn Malaysia, Batu Pahat 86400, Johor, Malaysia; 5ARL Laboratory Services PTY Ltd., 1361/55 Pine Rd., Yennora, Sydney, NSW 216, Australia; 6Faculty of Chemical and Energy Engineering, Universiti Teknologi Malaysia, Johor Bahru 81310, Johor, Malaysia; 7Advanced Engineering Materials and Composites Research Center (AEMC), Faculty of Engineering, Universiti Putra Malaysia, Serdang 43400, Selangor, Malaysia

**Keywords:** wheat starch, plasticizers, biocomposite film, properties improvement

## Abstract

Biocomposite materials are essential for environmental protection, as they have the ability of substituting synthetic plastic with natural materials. This work investigated how different plasticizers (Glycerol (G), Fructose (F), Sorbitol (S), and Urea (U)) affect the morphological, mechanical, thermal, and physical characteristics of films made of wheat starch at various concentrations (0%, 15%, 25%, and 35%). Plasticizers were added to improve the flexibility and homogeneity of the wheat starch-based bioplastic. Control film exhibited high tensile strength (38.7 MPa) with low elongation (1.9%). However, films plasticized with 35% sorbitol showed the highest elongation, which was 60.7% at break. At 35% of all plasticizers, fructose showed the highest tensile strength, with 7.6 MPa. The addition of different plasticizers shows improvement in water resistance; films plasticized with glycerol had the lowest water absorption at 35% fructose (187.4%) and also showed coherent surfaces. Glycerol, sorbitol, and urea films showed a higher mass loss compared to fructose films. Fructose showed the highest performance after the analysis of the results, with low water absorption, water content, and mass loss and with high mechanical performance at 35% of fructose. SEM images show that the addition of fructose and glycerol improves the surface homogenate, while sorbitol and urea have a less compact structure with large pores.

## 1. Introduction

Plastic substances are created by carbon atoms from a petroleum source [1]. These are non-degradable substances that poison both humans and animals; bacteria cannot break the chains of the synthetic polymers. According to the United Nations Environmental Program (UNEP) [2], synthetic plastic production costs the world economy $75 billion each year in natural capital. Plastic incineration and marine pollution are included in this cost, but it is also estimated that fully 30% of this number is the result of greenhouse gas emissions from raw material extraction and processing.

Biocomposite materials represent a promising alternative to petroleum-based plastic materials. Biocomposite materials have many advantages, such as low cost, light weight, availability, renewability, being environmentally friendly, recyclability and degradability [3,4,5,6]. Biocomposite materials are involved in many industrial applications, such as packaging applications, the food industry, marine railings, electronic components, devices for medical applications and automobile components [7,8,9,10]. Starch-based bioplastics account for 85–90% of those on the market and are made from native or modified starch, alone or combined with synthetic chemicals (Ashok et al., 2018). In general, starch has a high amylopectin percentage (up to 80%) and a low amylose content [11].

Wheat is a non-wood plant [12] from the Triticum family; the wheat plant is cultivated in countries worldwide. The total production of wheat in 2018–2019 was 764.4 million metric tons [13]; this amount increases year by year. Starch is a type of carbohydrate that makes up about 60–75% of wheat grains [14]. Starches are a good choice to make biopolymers films due to their high ability for decomposition and their ability to act as a matrix with different types of fillers. They have other advantages such as availability, low cost [15] and the ability to improve the properties of other materials. Due to the mechanical and physical properties, wheat starch, gluten and fiber are vital in the biopolymer industry. Glycerol as a plasticizer considerably increased the elongation and water vapor permeability of wheat starch-based films [16].

Plasticizers have the ability to increase the free volume between polymer chains when they are added to a system of polymers, allowing the chain segments to move and rotate more freely and enabling increased movement of polymer chains relative to one another. Therefore, a plasticized polymer would be less elastic and would deform with less force compared to ones that do not contain a plasticizer [17]. To minimize the strong interaction of hydrogen bonds within amylose and amylopectin molecules in starch, plasticizers have been added to the starch matrix to enhance the flexibility and reduce the stiffness of the films [18,19]. Different types of plasticizers have been used with starches, such as glycerol [20,21], sorbitol [22], fructose, urea, xylitol [23], tri-ethylene glycol, sucrose and triethanolamine [24,25]. J. Tarique et al. [26] reported that different plasticizer concentrations had an impact on the physical, structural, mechanical, thermal and barrier characteristics of plasticized films; the inclusion of plasticizers improves the performance of starch films as a whole. Similar findings were recently reported by other researchers [27].

Plasticizers are used in composite material to improve the flexibility and applicability of the starch-based biocomposite. Starch cannot be melted in its native form due to the strong hydrogen bonds; plasticizer molecules penetrate starch granules and reduce hydrogen bonds in high temperature, pressure and shear stress. Ultrasonic treatment refers to sound waves beyond the audible frequency range. In composite materials, ultrasonic treatment is used to alter the average molecular dispersion of natural and synthetic polymers in an aqueous solution to remove bubbles [28]. This paper aims to study the effect of various plasticizers (glycerol, fructose, sorbitol and urea) in different concentrations (0%, 15%, 25% and 35%) on the physical, mechanical, thermal and morphological properties of wheat starch-based films.

## 2. Materials and Methods

### 2.1. Material

Wheat starch was bought from a local supplier in Kajang, Selangor, Malaysia. The wheat starch contained 83.5% carbohydrate, 0.7% protein and 0.2% total fat. The amylose and amylopectin percentage was measured using the standard method described by Michael et al. [29] using a spectrophotometer device. Amylose and amylopectin content were 21.91% and 78.09%, respectively. It was found that 84.5% of the scattered particles were less than 212 μm in size. The water content of wheat starch was 0.895%, and the density was 1.6 g/cm^3^. The glycerol, fructose, sorbitol and urea were supplied by Evergreen Engineering & Resources SDN-BHD, Malaysia.

### 2.2. Film Preparation

A total of 5 g of isolated wheat starch was weighed, the starch was mixed with 100 mL of distilled water, and then the mixture was put in a water bath at 90 ± 5 °C with constant stirring for 20 min. After ensuring that the starch particles were mixed and the mixture took a slurry shape, the plasticizer was added to the slurry, and stirring was kept constant for another 20 min. The plasticizers were glycerol, fructose, sorbitol and urea, and the plasticizers were added in 15%, 25% and 35% *w*/*w*. Then, the slurry was treated with an ultrasonic device to remove bubbles and to improve the adhesion between particles. A total of 22 g from the treated slurry was cast (at room temperature) in Petri dishes (90 mm diameter). Then, the casted films were placed in a dryer oven at different times, as shown in Table 1. The formed films were peeled out of the casting plates and stored in plastic bags at room temperature for a week prior to the characterization processes. Figure 1 shows the produced films.

### 2.3. Physical Properties

#### 2.3.1. Film Thickness

The film thicknesses were determined according to the method of Lu et al. [30]; the thickness was measured by a micrometer, and the thickness was calculated from the average of five different specimens.

#### 2.3.2. Film Density

The density was measured for the square dimension (2 cm × 2 cm), and the volume was directly calculated by multiplying the thickness with the square area. The weight was measured by a laboratory sensitive balance, and the density was represented by calculating the average of three replicates. The density is a result of dividing the film weight by the film volume, following Equation (1).
(1)$$\rho =\frac{m}{v}=g/{\mathrm{cm}}^{3}$$
where *ρ* represents the density, *m* is the mass and *v* is the volume.

#### 2.3.3. Moisture Content (MC)

Moisture content was measured by determining the weight loss of the film before and after drying the specimens at 90 °C for 24 h; the moisture content was measured for (3 cm × 1 cm) for each sample, and the moisture content was estimated using the average of three replicates of the weight differences before (*m*_1_) and after (*m*_2_) dehydration according to Equation (2).
(2)$$MC=\frac{m_{1}-m_{2}}{m_{1}}\times 100\%$$

#### 2.3.4. Water Solubility (WS)

The water solubility test was performed by drying the samples at 90 °C for 24 h and then weighting their initial dry weight (*w_i_*), then the samples were soaked for 6 h in 50 mL of distilled water. Then, the specimens were dried at 90 °C for 24 h and then weighted their final weight (*w_f_*). Water solubility was calculated using Equation (3).
(3)$$WS=\frac{w_{i}-w_{f}}{w_{i}}\times 100\%$$

#### 2.3.5. Water Absorption (WA)

The water absorption test was conducted according ASTMD-570-98 standard; the film specimens were dried for 24 h at 50 °C, cooled in a desiccator, then directly weighed (*w_i_*). Then, the samples were immersed in distilled water, and the weight (*w_f_*) was measured each 30 min until the weight of the samples is settled. Water absorption was calculated according to Equation (4).
(4)$$WA=\frac{w_{f}-w_{i}}{w_{i}}\times 100\%$$

### 2.4. Film Transparency

The opacity of each film was measured to determine the transparency of each film. A spectrophotometer (BECKMAN COULTER/DU 730, Beckman Coulter, Brea, CA, USA) was used to measure the transparency according to Equation (5).
(5)$$\mathrm{Opacity}=\frac{\mathrm{Abs}600}{x}$$
where *x* represents the thickness (in mm) of the film, and Abs600 is the absorbance of light measured at 600 nm.

### 2.5. Biodegradation of Biocomposites (Soil Burial Test)

The biodegradation test was conducted using the method described by La Mantia et al. [31]. The specimens were dried and weighted (*w_i_*) before being buried in 5 cm of soil in a confined setting (plastic boxes). Weight loss (*WL*) was measured in triplicate by taking samples at different times and cleaning them with a brush. They were then dehydrated for 6 h at 105 °C and weighed (*W_f_*). The degradation test was acquired on each of the two days and calculated using Equation (6):(6)$$WL=\frac{w_{i}-w_{f}}{w_{i}}\times 100\%$$

### 2.6. Simultaneous Thermal Analysis (STA)

The STA test was conducted using an analyzer (NETZSCH STA 449F3, Selb, Germany). Under a nitrogen environment, 10 mm^2^ films were deposited in platinum crucibles and heated at a steady rate of 10 °C/min from ambient temperature to 500 °C. The TGA curve was used to explain the thermal stability of samples and to assess mass loss over time as a function of temperature in this method of thermal analysis. It also described the decomposition of different components of the composite in DTG curves.

### 2.7. Structural Properties

#### 2.7.1. Scanning Electron Microscopy (SEM)

The type of the device that used in this test was (VPSEM CARL ZEISS EVO MA 10, Jena, Germany), to investigate the surface morphology of the specimens. Before delivering a 20 kV acceleration voltage through a high vacuum, each specimen was coated with a golden layer. This test yielded high-resolution photos at various magnification settings.

#### 2.7.2. Fourier Transform Infrared Spectroscopy (FTIR)

An IR spectrometer (Bruker Vector 22 IR spectrometer, Bruker, USA) was used to determine the FTIR spectra using the frequency range of 4000–650 cm^−1^. The test was conducted using the FTIR-ATr method.

#### 2.7.3. X-ray Diffraction (XRD)

The X-ray diffraction analysis was measured by an X-ray diffractometer (D8 ADVANCE | Bruker, Rigaku-Tokyo, Japan). The device was managed by a 0.02 (θ) s^−1^ scattering speed within a 5° to 60° (2θ) angular range under an operating voltage and current of 40 kV and 35 mA, respectively, according to Equation (7).
(7)$$Ci=\frac{A_{c}}{A_{c}+A_{a}}\times 100\%$$
where *Ci* represents the crystallinity index, *A_c_* is the crystallinity area and *A_a_* is the amorphous area in the XRD pattern.

### 2.8. Water Vapor Permeability (WVP)

Water vapor transmission rate (*WVTR*) was measured according to standard ASTM E96-00. Before starting the test of water vapor permeability, the film was subjected for 48 h to 25 °C and 67% relative humidity; samples were placed over than mouth of the test cup, and the cup was prefilled with anhydrous calcium chloride. Then, the sample was placed under conditions of 25 °C and 75% relative humidity. Water vapor permeability (*WVP*) was calculated according Equations (8)–(10):(8)$$WVP=\frac{m\times t}{A\times T\times P}$$
(9)$$WVTR=\frac{m}{A\times T}$$
(10)$$WVP=\frac{WVTR}{P\left(R_{1}-R_{2}\right)}\times t$$
where *WVP* is water vapor permeability, *WVTR* is water vapor transmission rate, *t* is film thickness (m), *m* is the weight increment of the cup (g), *A* is the area exposed (m^2^), *T* is the time lag for permeation (s), *P* is water vapor partial pressure difference across the film (Pa), *R_1_* is the RH in the desiccator, *R*_2_ is the RH in the cup and *t* is the film thickness (m).

### 2.9. Tensile Test

The mechanical properties of the films (tensile strength, tensile modulus, and elongation) were determined using the ASTM-D882 standard. Film strips were cut into 70 × 10 mm^2^ sections and then characterized using a (SHIMADZU/KUA 0400MED 00693, Kyoto, Japan) tensile machine using a 500 N load with an initial grip separation and crosshead speeds of 30 mm and 1 mm/min, respectively. At least four replicates were carried out for each sample.

## 3. Results and Discussion

### 3.1. Physical Properties

#### 3.1.1. Thickness and Density

There was no remarkable difference between thickness and density in different films. Adding more plasticizers increased the thickness of the film, as shown in Table 1, while the density decreased slightly with the addition of the plasticizers. The increment in the thickness of the films after adding plasticizers can be attributed to readily and easily permeating the starch network by the plasticizers [25]. The highest thickness was obtained from 35% urea, while the lowest thickness was obtained from 15% sorbitol. The highest density was 1.59 g/cm^3^ with 15% fructose, while the lowest density was 1.07 g/cm^3^ with 35% urea.

#### 3.1.2. Moisture Content and Water Solubility

Moisture content and water solubility are important properties for applications that require a specific amount of moisture content and specific insolubility. The addition of plasticizers increases the water content gradually. Similar results were reported by Asgar by using glycerol as a plasticizer [32]. The increment of water solubility is ascribed to the plasticizer’s effects. Due to its hydrophilic nature, plasticizers minimize the interactions between biocomposite particles and promote solubility, providing the polymer particles with a higher propensity to bind water. Similar results were reported by Ghasemlou et al. [33]. The highest amount of moisture content was 21.53%, which was recorded for 35% urea, while the lowest amount of moisture content was 8.17%, which was recorded for 25% fructose. The highest amount of water solubility was 29.02%, which was recorded for 35% fructose, while the lowest amount of water solubility was 2.54%, which was recorded for 0% plasticizers (the control films).

#### 3.1.3. Water Absorption

Water acts like a plasticizer by improving the flexibility of the material [34]. Nevertheless, water absorption is not a desirable property, because of the change of the dimensions; swelling is a result of water particle absorption. The hydrophilic nature of a plasticized film decreases with the addition of plasticizer content, which means higher water content in the film leads to increased water saturation. Hydroxyl groups in starch and plasticizer particles are the reason for the high water absorption of the plasticized polymer [35]. Plasticizers can combine with starch by producing hydrogen bonds. Plasticizers’ OH groups can establish hydrogen bonds with starch. The water absorption is consistent with the water solubility and water content.

Water absorption decreased with the increment of the plasticizer percentage, as shown in Figure 2, which is ascribed to the properties of the plasticizers; they are less soluble in water, and they have a less hydrophilic nature than starch. The control film absorbs about 450% in 360 min, whereas the plasticized films absorbed less than that. After 360 min, the weight of the immersed films settled down because of the saturation of the film. Films plasticized with fructose give the lowest water absorption, at 35% glycerol (140.1%), which makes it the most water resistant, while at the same percentage of plasticizer, film plasticized with fructose, sorbitol and urea absorbed 187.4%, 234% and 300.7% respectively.

### 3.2. Film Transparency

The importance of film transparency has different evaluations depending on the application; however, film transparency is highly desired in food packaging applications. Opacity is the opposite of film transparency, where the film thickness is considered. The effect of various plasticizers shows different results; the addition of fructose tends to reduce the opacity at 25% and 35%, with 0.620 and 0.6246 nm/mm, respectively, which was the lowest opacity for all samples. Plasticizing films with fructose, glycerol and sorbitol with 15% load showed an opacity higher than 25% and 35% of the plasticizer. On the other hand, the urea sample revealed a higher opacity, especially at 35%. Table 2 shows the transparency of wheat starch-based films and various plasticizers.

Daniel et al. [36] observed that the type of starch also affects the opacity; their results were similar to our work on plasticized wheat starch (0.93 nm/mm); this amount of opacity is considered as the second lowest opacity after potato starch-based films.

### 3.3. Soil Degradation

Burying the samples in moist soil caused enzymatic degradation. Figure 3 shows the degradation percent every two days; it was found that the addition of the different plasticizers reduces the degradation. Except for urea films, which showed the fastest decomposition, 35%U fully degraded within 8 days. It was difficult to handle films plasticized with urea because of the fragmentation that occurred after one week; this fragmentation also happened with the other films after 10 days. The degraded mass increased until it reached 0% within two weeks. Many bacteria and fungi in the soil environment can break down starch’s primary ingredients (amylose and amylopectin). Microorganisms can attack the carbohydrate structure and have particular enzymes that can hydrolyze these polymers into digestible units [37]. From the result in Figure 3, it is clear that the addition of a plasticizer reduces the acceleration of the degradation, which means that the plasticizer creates stronger bonds and improves particles adhesion.

### 3.4. Water Vapor Permeability

To develop useful and effective bioplastic, the transformation of the moisture between the plastic sides needs to be reduced [38,39]. Table 1 shows that the addition of the plasticizer increases the WVP; similar results were recorded by Muhammed L. Sanyang et al. [19] and Muscat et al. [40]. Films plasticized with urea showed the highest WVP; 35% urea revealed 1.88133 × 10^−10^ g·mm·s^−1^ m^−2^ Pa^−1^, while the addition of glycerol showed the lowest WVP, starting with 0.907385 × 10^−10^ g·mm·s^−1^ m^−2^ Pa^−1^ for 15% glycerol until 1.387 × 10^−10^ g·mm·s^−1^ m^−2^ Pa^−1^ for 35% glycerol.

The hydrophilic nature of biopolymers is the main reason for their WVP, which encourages the sorption of water molecules [41,42]. Low moisture content causes starch–water interactions to form stronger hydrogen bonds than starch–starch intermolecular chains. The absorbed water diffusion behavior through the matrix is changed by an increase in free volume inside the polymeric matrix by adding plasticizers. As a result, the polymer networks become less thick, facilitating the adsorption of water molecules on the film’s surface (increasing solubility) and easier penetration through its structure (higher diffusivity), resulting in higher WVP [43].

### 3.5. Structural Properties

#### 3.5.1. Scanning Electron Microscopy (SEM)

SEM testing was conducted to show images under 300× magnification; the images show that urea film surfaces are coarse and covered with impurities and agglomerates of non-melted starch. The addition of other plasticizers showed fewer pores and cracks; 35% plasticizer showed the smoothest surface, especially with the fructose. This was attributed to the fact that a high amount of plasticizer showed smoother surfaces [44]; a homogenous surface can indicate high tensile properties [25]. A total of 15% of plasticizers showed less smooth surfaces, with large porosity as well as micro-cracks in the structure. The control film and sorbitol addition also showed a smooth surface with some cracks. Figure 4 shows SEM images of wheat starch-based films and various plasticizers.

#### 3.5.2. Fourier Transform Infrared (FT-IR) Spectroscopy

The interaction between the particles of wheat starch and plasticizers was investigated with an FT-IR spectroscope; the FTIR spectrum curve was classified into four primary sections: the first section had wavenumbers less than 800 cm^−1^, the second section between 800 and 1500 cm^−1^ wavenumbers, the third section ranged from 2800 to 3000 cm^−1^, and the last section had wavenumbers above 3000 cm^−1^ [45]. Because the elemental makeup of the plasticized films was based on the starch structure, the curves were essentially identical, as shown in Figure 5.

The control film showed peaks at 3280.07 cm^−1^, 2924.46 cm^−1^, 1635.43 cm^−1^, 1337.94 cm^−1^, 1149.45 cm^−1^, 1076.98 cm^−1^, 994.13 cm^−1^, 929.88 cm^−1^, 859.23 cm^−1^ and 759.70 cm^−1^. The peak observed at 3280.07 cm^−1^ was ascribed to stretching of the O-H groups [46], the 2924.46 cm^−1^ sharp peak attributed to the C-H group [47], the 1635.43 cm^−1^peak referred to bending mode of the absorbed water, the 1337.94 cm^−1^ peak was a sign for CH_2_ bonding [48], the 1149.45 cm^−1^ sharp peak was a sign for the coupling of C-C and C-O stretching mode [49], the 1076.98 peak referred to C-O-H group in wheat starch [50], the 994.13 cm^−1^ peak was a sign of C-O stretching in C-O-C and C-O-H in the glycosidic ring of starch [51], the 860.35 cm^−1^ peak indicated strong C–H bending [52] and the 759.70 cm^−1^ peak indicated vibrations of the glucose pyranose unit [45].

There is no significant difference between the spectrum of the control and plasticized films, except for a slight shift in stretching of the O-H group; the slight shift occurred in a higher wavenumber in glycerol films and in a lower wavenumber in fructose and sorbitol films.

#### 3.5.3. X-ray Diffraction (XRD)

To study the effect of different plasticizers in various concentrations on the crystallinity index, an XRD test was conducted. Figure 6 shows the crystalline peaks of plasticized wheat starch films. It shows that plasticized films have a lower crystallinity than the control film. The amount and type of plasticizer did not significantly affect the crystallinity of the films; after making the comparison, all plasticized films had a lower crystalline index than the control film. The plasticizers showed that the urea and fructose films had a higher crystallinity than the sorbitol and glycerol films, as shown in Table 1.

The framework of the starch was destroyed due to gelatinization and retrogradation. Figure 6 shows that the control film had sharp diffraction peaks located at 12.36°, 14.24°, 15.08°, 17.32°, 19.96° and 21.6°. The fructose plasticized film showed a similar pattern to the control film, except that a new peak appeared at 34.6° for the 25% fructose, and the peak at 17.32° seemed to be sharp when adding the fructose. Plasticizing film with glycerol and sorbitol gave a similar pattern to the control film, except that for sorbitol plasticized film, the peak at 15.08° was more intense when the concentration of the plasticizer increased. Films plasticized with urea revealed a slight shift to a higher degree on all peaks, while the peak at 19.96° was more intense with the addition of urea.

### 3.6. Simultaneous Thermal Analysis (STA)

The STA test was used as an analytical technique to measure the thermal stability and its fraction of volatile components by monitoring the weight change that occurs as a sample is heated at a predetermined heating rate [53,54]. Plasticized wheat starch films were analyzed using a thermal gravimetric analyzer device to compare the thermal properties of different plasticized films in various percentages.

Based on Figure 7 illustrating the TGA curves, glycerol, sorbitol and urea films showed a higher mass loss compared to fructose films. The low water content of fructose films, as stated in Table 1, is most likely attributed to this observation. This led us to conclude that the higher the water content, the higher the mass loss in the phase of losing water mass.

The derivative thermogravimetric (DTG) curves show three distinctive phases of the degradation process. The initial mass loss was produced by a thermal drop below 125 °C, which was mostly due to the elimination of moisture and water fragments via evaporation [55,56]. The second phase began with further heating at temperatures ranging from 150 °C to 250 °C. This mass loss can be attributed to the volatilization of plasticizer molecules in the presence of water [57]. Sorbitol and glycerol did not show a separate phase for their volatilization; this refers to their low volatilization temperature [58], which makes them volatile after losing water content. At temperatures above 270 °C, the last step of film degradation’s mass loss was revealed. Weight loss was connected to the depolymerization and breakdown of carbon chains in the starch structure during this time. The onset of the thermal reactions of starches is widely known to occur at around 300 °C. [34]. According to the data in Figure 8, all films decompose at temperatures ranging from 298.6 °C to 312.4 °C; the lowest degradation temperature was recorded for the control film (298.6 °C), and the highest temperature recorded for 15% glycerol (312.4 °C). These findings show that adding plasticizers to wheat starch-based films did not significantly alter their thermal stability. From Figure 9, it is obvious that fructose and urea films have the highest mass residues at 495 °C. The addition of 15% fructose resulted in the highest mass residues at 495 °C (25.66%), while 25% sorbitol had the lowest (6.7%).

### 3.7. Tensile Properties

The mechanical properties measured in the tensile test were tensile modulus, strength and elongation. Based on the results in Figure 10, it is obvious that the increment in plasticizer content reduces the tensile strength and modulus while it improves the elongation. As seen in Figure 10, the control film gives the highest tensile and modulus, with values of 37.81 MPa and 1629.8 MPa, respectively, while the lowest elongation was recorded at break (1.94%). For the plasticized films, the highest tensile was 25.3 MPa, which recorded for 15% sorbitol; the tensile strength dropped until it reached 6.35 MPa with 35% sorbitol. Film plasticized with glycerol showed the second highest tensile strength of 22.8 MPa for 15% glycerol, and with the addition of glycerol content it reached 6.35 MPa. Films plasticized with fructose showed the third highest tensile strength, 18.99 MPa for 15% fructose; the addition of fructose did not reduce the tensile strength like the other plasticizers, as the tensile strength was 12 MPa for 25% fructose and 7.6 for 35% fructose. Films plasticized with urea showed the lowest tensile strength, starting with 5.64 MPa for 15% urea and ending with 1.12 MPa for 35% urea.

The modulus shows the same trend of tensile strength in terms of the reduction after adding the plasticizers. At 15% of the plasticizer load, fructose, glycerol, sorbitol and urea were recorded at 822.22 MPa, 1119 MPa, 1230.355 MPa and 224.14 MPa, respectively. When 25% was applied, fructose, glycerol, sorbitol and urea were recorded at 435.83 MPa, 199.19 MPa, 310.23 MPa and 12.367 MPa, respectively. At the final load of 35% of the plasticizer, fructose, glycerol, sorbitol and urea were recorded at 152.54 MPa, 68.76 MPa, 121 MPa and 6.56 MPa, respectively. Figure 11 shows the decrease in stress with the addition of plasticizer content, while the flexibility of the film is improved.

Because of its capacity to hydrolyze the molecular link structure of starch when heated together, water is frequently utilized as the main plasticizer [59]. The main reason for adding further plasticizers is to improve the flexibility; elongation at break of the samples represents the flexibility. Films plasticized with 35% sorbitol show the highest elongation, which was 60.7% at break, while the addition of sorbitol at 15% and 25% gives 3.3% and 34.1%, respectively. The addition of glycerol also improves the elongation at break; 15%, 25% and 35% of glycerol give 4.4%, 46.1% and 47.2%, respectively. Urea addition gives high elongation at the lowest percentage (15% of urea gives 32.3%); this improvement does not continue rapidly with the addition of urea, as 25% and 35% urea gives 50.4% and 54.1% respectively. Fructose addition at 15%, 25% and 35% gives 9.41%, 17.2% and 47.5% elongation at break, respectively. Although, fructose samples did not give the highest elongation at break, it was noted that the addition of fructose did not reduce tensile properties like other plasticizers.

Comparing this result with other starch-based biocomposites such as Jackfruit seed starch [60] and potato starch [61] reveals that plasticizing starch with a load of (15–35%) shows good tensile strength while maintaining acceptable flexibility. However, the effect of ultrasound treatment on the mechanical properties needs more investigation, especially for the aspect of the molecular structure.

## 4. Conclusions

In this work, we investigated the effect of various plasticizers (glycerol, fructose, sorbitol and urea) in different concentrations (0%, 15%, 25% and 35%) on the physical, mechanical, thermal and morphological properties of wheat starch-based films. The findings show a significant improvement in the flexibility with all plasticizers; however, 35% gives the highest elongation for different samples. Films plasticized with 35% sorbitol show the highest elongation, which was 60.7% at break, and the control film exhibited the highest tensile strength (38.7 MPa) with low elongation (1.9%). At 35% of all plasticizers, fructose shows the highest tensile strength with 7.6 MPa. It was observed that urea films show high elongation at 15% of urea. We recommend further investigation of the effect of urea in loads less than 15%. The addition of different plasticizers shows improvement in water resistance; films plasticized with glycerol give the lowest water absorption at 35% glycerol (141%). The addition of different plasticizers shows smooth and coherent surfaces in morphological scans. Glycerol, sorbitol and urea films showed a higher mass loss compared to fructose films. SEM images show that the addition of fructose and glycerol improves the surface homogenate, while sorbitol and urea have a less compact structure with large pores. Overall, 35% fructose shows the highest performance after the analysis of the results, with low water absorption, water content, and mass loss and with a high mechanical performance at 35% of fructose.

## Figures and Tables

**Figure 1 polymers-15-00063-f001:**
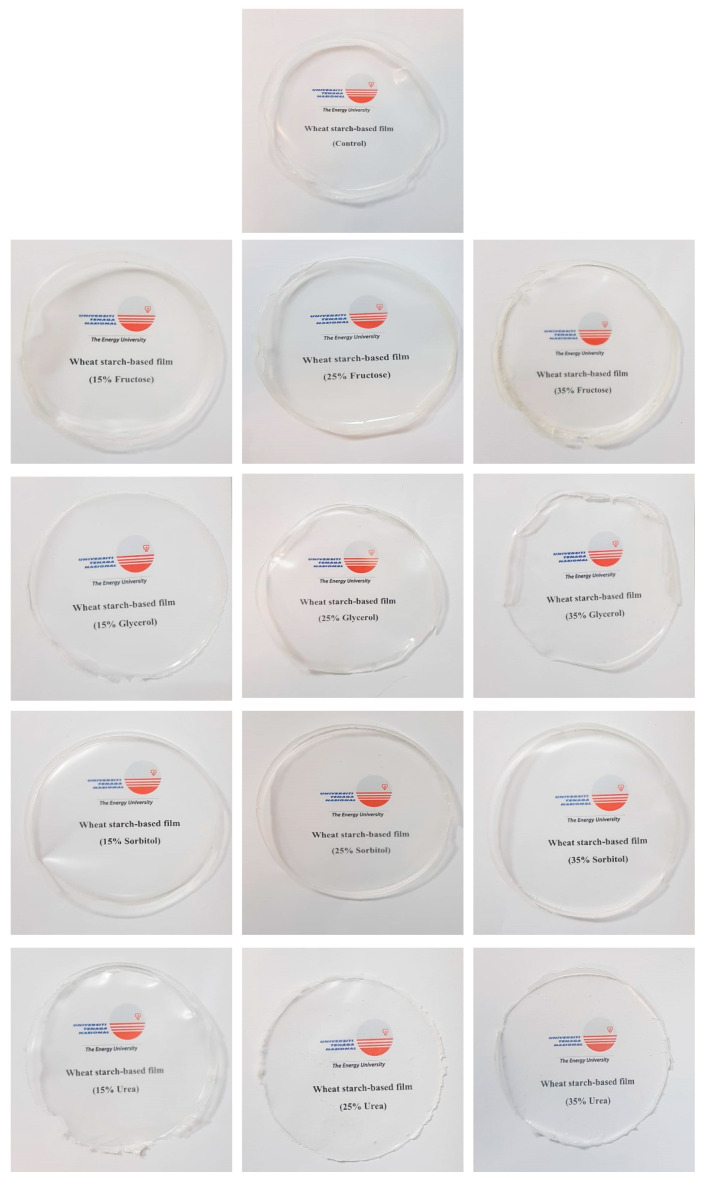
Different plasticized films produced.

**Figure 2 polymers-15-00063-f002:**
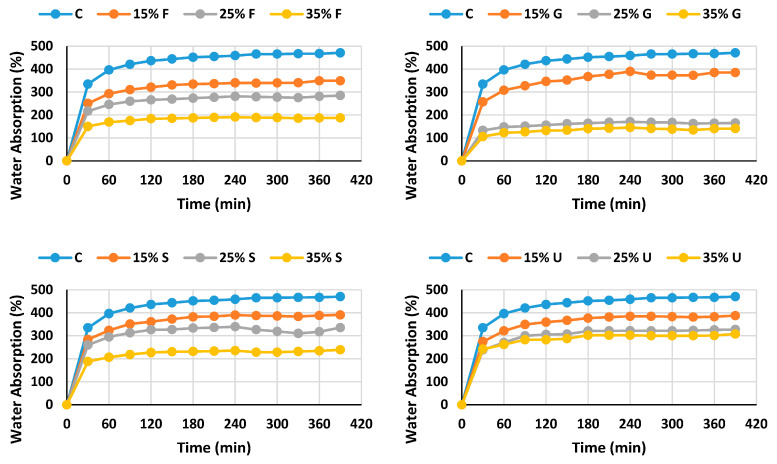
Water absorption of wheat starch-based films and various plasticizers.

**Figure 3 polymers-15-00063-f003:**
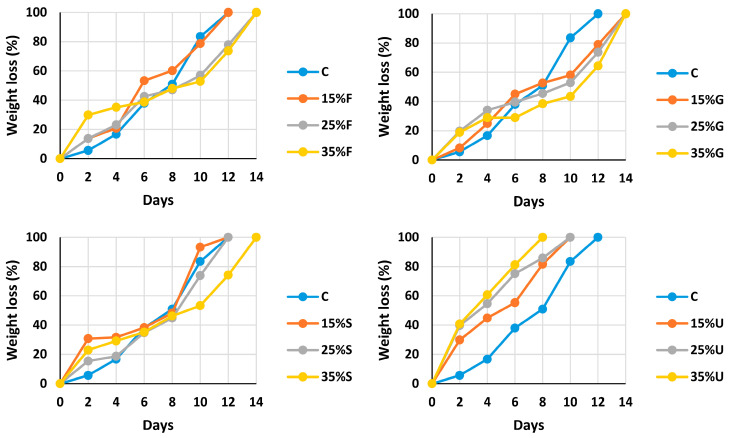
Degradation of wheat starch films with various plasticizer types at different concentrations.

**Figure 4 polymers-15-00063-f004:**
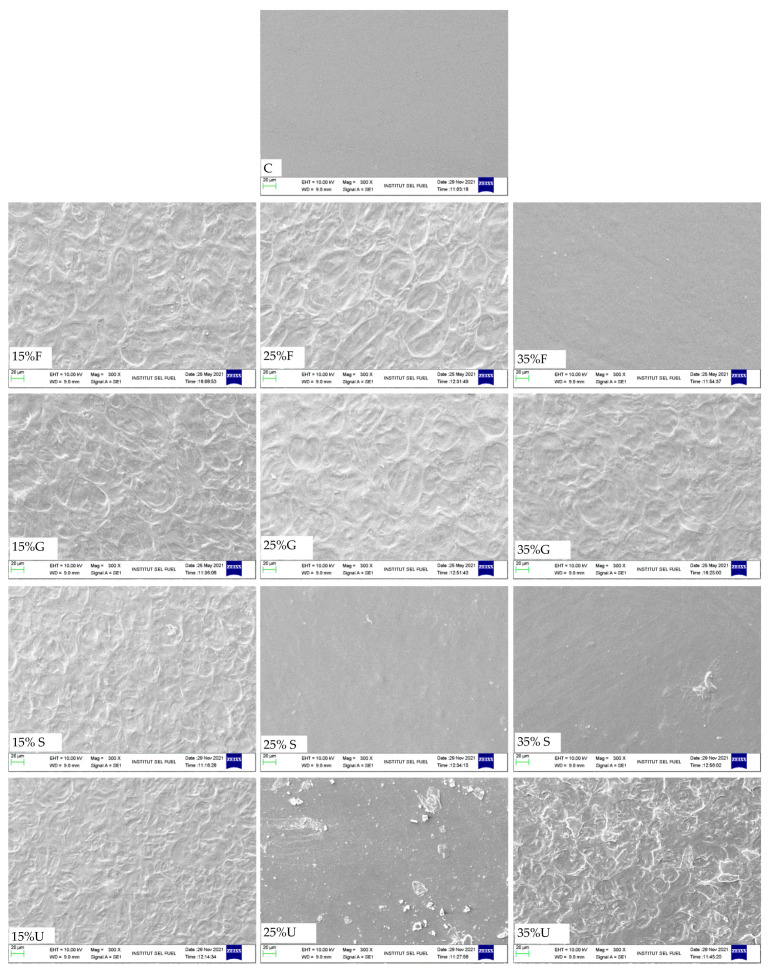
SEM images of wheat starch films with various plasticizer types at different concentrations.

**Figure 5 polymers-15-00063-f005:**
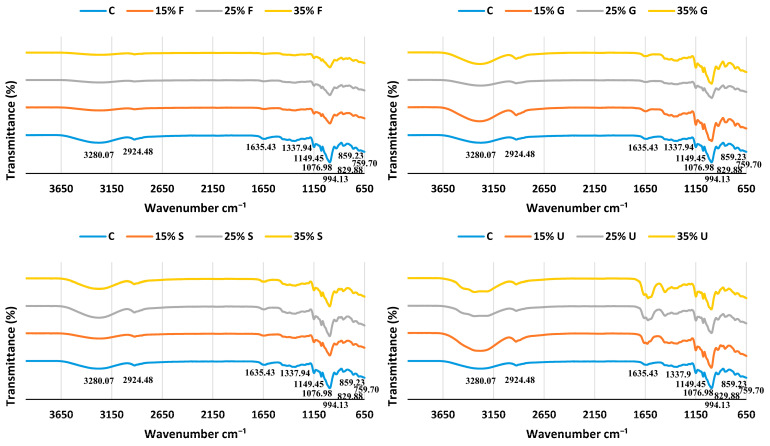
FTIR curves of wheat starch films with various plasticizer types at different concentrations.

**Figure 6 polymers-15-00063-f006:**
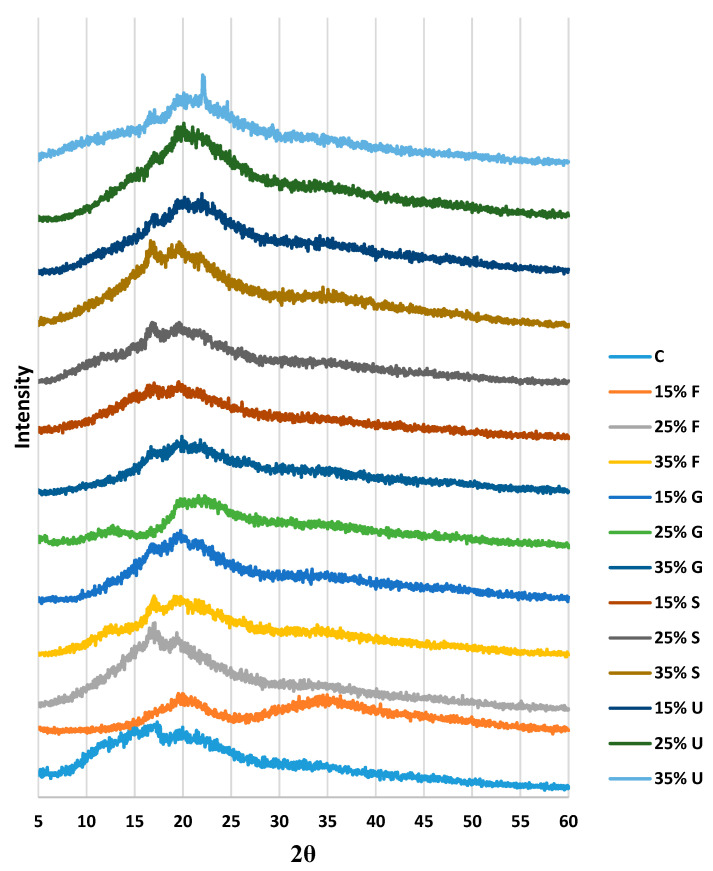
XRD curves of wheat starch films with various plasticizer types at different concentrations.

**Figure 7 polymers-15-00063-f007:**
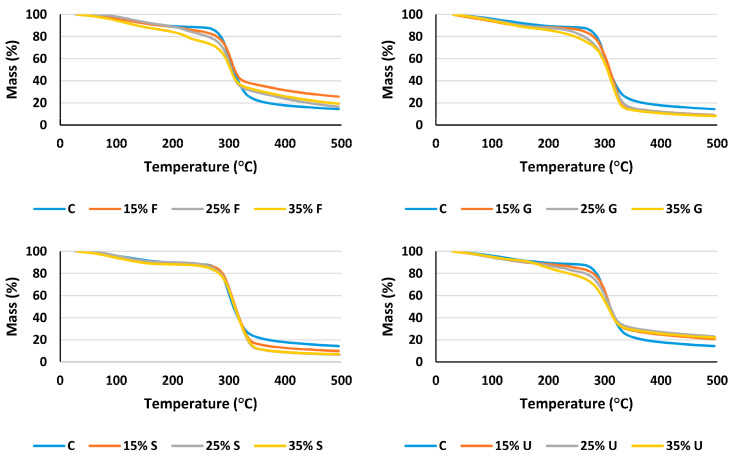
TGA curve of wheat starch plasticized films with various plasticizer types and concentrations.

**Figure 8 polymers-15-00063-f008:**
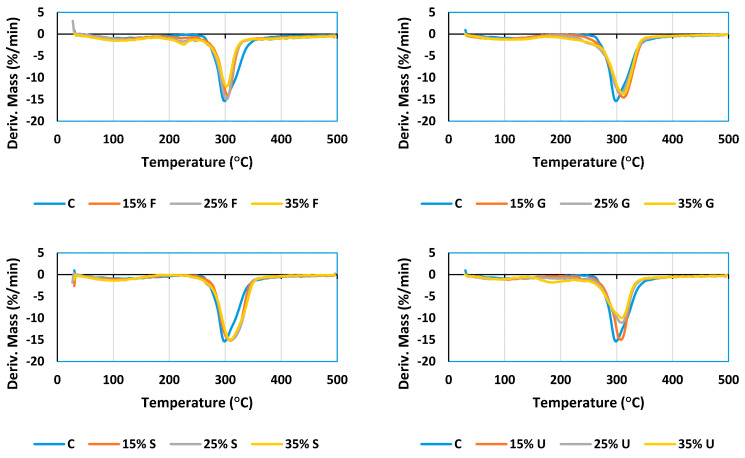
DTG curve of wheat starch plasticized films with various plasticizer types and concentrations.

**Figure 9 polymers-15-00063-f009:**
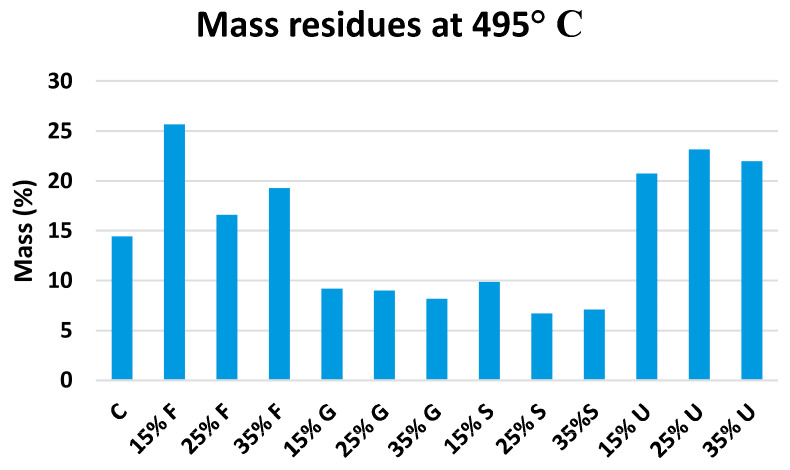
Mass residues of plasticized wheat starch films at 495 °C.

**Figure 10 polymers-15-00063-f010:**
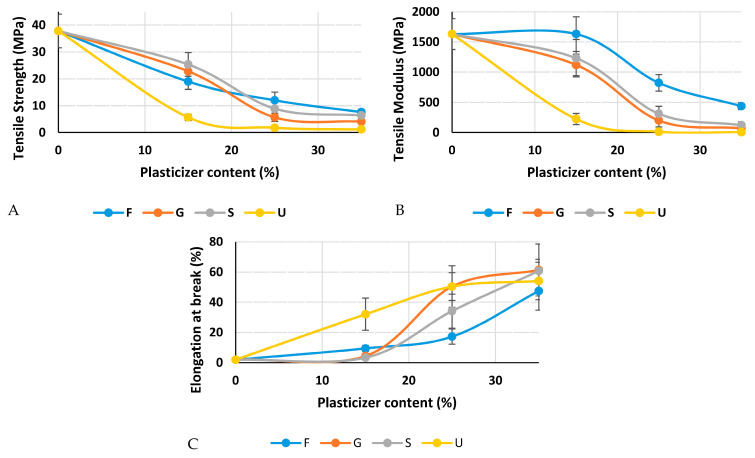
Mechanical properties of wheat starch plasticized films. (**A**) Tensile Strength, (**B**) Tensile Modulus and (**C**) Elongation at break. 0% is the control film.

**Figure 11 polymers-15-00063-f011:**
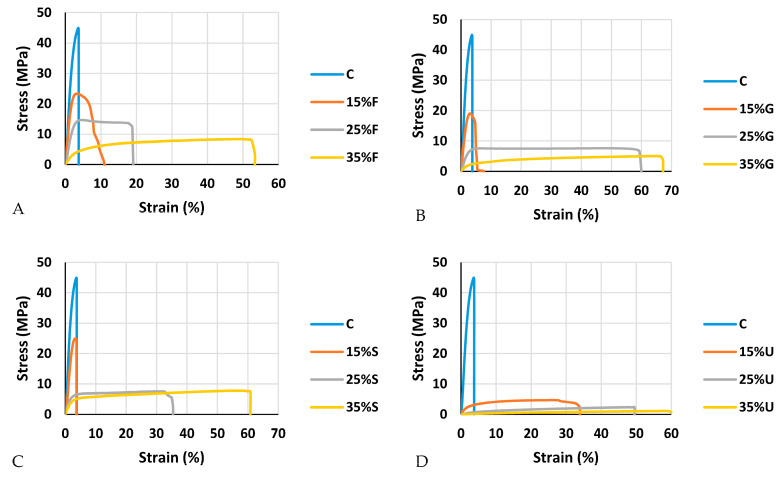
Stress–stain curves of wheat starch-based film with different plasticizers. (**A**) Fructose, (**B**) Glycerol, (**C**) Sorbitol and (**D**) Urea.

**Table 1 polymers-15-00063-t001:** Physical properties, *WVP*, crystallinity and drying time of wheat starch-based films and various plasticizers.

Film	Thickness, µm	Density, g/cm^3^	Water Content, %	Water Solubility, %	*WVP,* 10^−10^ g·mm·s^−1^ m^−2^ Pa^−1^	Crystallinity Index, %	Drying Time, hours
C	170.20	1.32	11.86	2.54	1.12	18.90	6.00
15% F	156.40	1.59	8.70	14.92	1.22	17.80	7.00
25% F	177.60	1.47	8.17	22.62	1.49	16.50	15.00
35% F	200.20	1.39	9.15	29.02	1.53	16.90	22.00
15% G	176.60	1.55	12.14	12.92	0.91	14.50	7.00
25% G	205.20	1.37	13.67	19.27	1.13	15.50	24.00
35% G	207.60	1.34	20.58	20.00	1.39	16.70	44.00
15% S	155.00	1.55	10.38	15.19	1.24	13.10	7.00
25% S	175.20	1.49	10.47	22.06	1.45	14.90	15.00
35% S	189.60	1.48	9.90	28.00	1.58	14.10	22.00
15% U	170.20	1.48	11.48	13.29	1.66	18.80	20.00
25% U	190.40	1.45	16.98	17.81	1.77	15.50	16.00
35% U	207.60	1.07	21.53	20.42	1.88	17.30	38.00

**Table 2 polymers-15-00063-t002:** Transparency of wheat starch-based films and various plasticizers.

Sample	Opacity (A600/mm)
C	0.808
15%F	1.032
25%F	0.620
35%F	0.624
15%G	0.926
25%G	0.723
35%G	0.655
15%S	0.954
25%S	0.615
35%S	0.585
15%U	0.766
25%U	0.774
35%U	1.806

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
