# Peer review of "Effect of Various Plasticizers in Different Concentrations on Physical, Thermal, Mechanical, and Structural Properties of Wheat Starch-Based Films"

_polymers, 2022, doi:10.3390/polym15010063_

Round 1
Reviewer 1 Report
1. Why different drying times are used in preparing films? Does this affect properties such as the moisture content of the film?
2. It is suggested to delete the role of " ultrasonic treatment " in the introduction. In addition, "2.10. Statistical analyses" should also be deleted.
3. It is recommended to explain the plasticizing mechanism of these plasticizers on starch film. The influence of different plasticizers on the performance of the film should also be analyzed in depth in combination with the plasticizing mechanism.
4. In "2.3.4. Water solubility (WS)", "w2 in moisture content is the initial weight in water solubility" is repeated and contradictory to "wi is the initial dry weight". Before the water solubility test, whether to dry the film to a constant weight?
5. Data processing and mapping capabilities need to be improved.
Author Response
Point 1: Why different drying times are used in preparing films? Does this affect properties such as the moisture content of the film?
Response 1:
Thank you for this point, we took the samples out of the dryer oven after we make sure that the weight of each sample is stable, and no change happed in the weight. That was done because the control films and films with low amount of plasticizers show stability in weight in less time. Moreover, when the films were left at the oven after they dry, cracks occurred in the samples with no change in the weight. To avoid the effect “the changes in the properties” we did all the tests after one week of the preparation day. This point has been added in lines 101- 102 “The formed films were peeled out of the casting plates and stored in plastic bags at room temperature for a week prior to characterization processes.”
Point 2: It is suggested to delete the role of " ultrasonic treatment " in the introduction. In addition, "2.10. Statistical analyses" should also be deleted.
Response 2:
Thank you for this important point. We removed both parts.
Point 3: It is recommended to explain the plasticizing mechanism of these plasticizers on starch film. The influence of different plasticizers on the performance of the film should also be analyzed in depth in combination with the plasticizing mechanism.
Response 3:
Thank you for this valuable comment, we added the explanation of the plasticizing mechanism in lines 64-68 “Plasticizers have the ability to increase the free volume between polymer chains when they are added to a system of polymers, allowing the chain segments to move and rotate more freely and enabling increased movement of polymer chains relative to one another. Therefore, a plasticized polymer would be less elastic and would deform with less force compared to the one which didn't contain a plasticizer.” And in lines 335- 341 “The absorbed water diffusion behavior through the matrix is changed by an increase in free volume inside the polymeric matrix by adding plasticizers. As a result, the polymer networks become less thick, facilitating the adsorption of water molecules on the film's surface (increasing solubility) and easier penetration through its structure (higher diffusivity), resulting in higher WVP [44].
Point 4: In "2.3.4. Water solubility (WS)", "w2 in moisture content is the initial weight in water solubility" is repeated and contradictory to "wi is the initial dry weight". Before the water solubility test, whether to dry the film to a constant weight?
Response 4:
Thank you for this valuable comment. The w2 in moisture content can represent wi in water solubility, as we performed the testing sequentially, w2 in moisture content and wi in water solubility both are the weight of the samples after drying it at 90°C for 24 h. But we removed the part “w2 in moisture content is the initial weight in water solubility” to avoid any contradiction. And we make it more clear by stating “Water solubility test performed by drying the samples at 90°C for 24 h and then soaking them for 6 hours in 50 ml of distilled water. after that the specimens have been dried at 90°C for 24 h and then weighted. Water solubility calculated using equations (3).
(3)
Where WS represents the water solubility, wi is the initial dry weight and wf is the final dry weight.”
Point 5: Data processing and mapping capabilities need to be improved.
Response 5: Thank you for this valuable point, we improved the mapping with more quality figures and organizing TGA and FTIR figures.

Reviewer 2 Report
1. The authors claimed that plasticizers can improve the surface roughness of thin films. AFM images should be provided to investigate the morphology.
2. In Figure 4, the scale bar is blur. And all words at the bottom of images are hard to read.
3. In section 3.2, it would be better if the author exhibit the effect of adding different plasticizers on the transparency of the film more visual.
4. In section 3.4, the influence of plasticizer on water vapor permeability could be provided more visually.
Author Response
Point 1: The authors claimed that plasticizers can improve the surface roughness of thin films. AFM images should be provided to investigate the morphology.
Response 1:
Thank you for this point, we corrected this information by showing more accurate claim “the images show that urea films surfaces are coarse and covered with some impurities and agglomerates of none melt starch.”, as these phenomena occurred with the urea samples, this finding agreed with other authors claims when they add urea as plasticizer to different types of starch “M.I.J. Ibrahim, S.M. Sapuan, E.S. Zainudin & M.Y.M. Zuhri (2019) Physical, thermal, morphological, and tensile properties of cornstarch-based films as affected by different plasticizers, International Journal of Food Properties, 22:1, 925-941, DOI: 10.1080/10942912.2019.1618324”.
Point 2: In Figure 4, the scale bar is blur. And all words at the bottom of images are hard to read.
Response 2:
Thank you for this important point. We have provided a google drive link for the editor that contain the original scale for the images, we will make sure that the images will appears very clearly with their description. We put them in this order to avoid using much space. https://drive.google.com/drive/folders/1MSqNLWB9PB-tFGPe7Im8kNt3StGaficX?usp=sharing.
Point 3: In section 3.2, it would be better if the author exhibit the effect of adding different plasticizers on the transparency of the film more visual.
Response 3:
Thank you for this valuable comment, the visualisation of the films is shown in figure 1, yet, there is no significance difference between the films.
Point 4: In section 3.4, the influence of plasticizer on water vapor permeability could be provided more visually.
Response 4:
Thank you for this valuable comment, we added those lines for more explanation in lines 335- 341 “The absorbed water diffusion behavior through the matrix is changed by an increase in free volume inside the polymeric matrix by adding plasticizers. As a result, the polymer networks become less thick, facilitating the adsorption of water molecules on the film's surface (increasing solubility) and easier penetration through its structure (higher diffusivity), resulting in higher WVP [46].

Reviewer 3 Report
The paper presented by the authors focuses on the Effect of various plasticizers in different concentrations on, thermo-physical -mechanical properties. In my opinion, the results of the experiments are good and high quality. In addition, the figures presented in the work are very clear. So I suggest to accepting it after minor revision.
-L23- please add "and" between 25% and 35%.
-Please replace '*' and 'X' in L113, L122, L125, 133, L159 and 185with the product symbol ×.
Author Response
Point 1: -L23- please add "and" between 25% and 35%.
-Please replace '*' and 'X' in L113, L122, L125, 133, L159 and 185with the product symbol ×.
Response 1:
Thank you for this important point, we replaced them with the correct symbol, and we added and between 25% and 35%.

Round 2
Reviewer 2 Report
All questions have been answered clearly. It can be published as present form.